# Immune Checkpoint Inhibitors in Urothelial Bladder Cancer: State of the Art and Future Perspectives

**DOI:** 10.3390/cancers13174411

**Published:** 2021-08-31

**Authors:** Giandomenico Roviello, Martina Catalano, Raffaella Santi, Valeria Emma Palmieri, Gianmarco Vannini, Ilaria Camilla Galli, Eleonora Buttitta, Donata Villari, Virginia Rossi, Gabriella Nesi

**Affiliations:** 1Department of Health Sciences, University of Florence, 50139 Florence, Italy; giandomenico.roviello@unifi.it (G.R.); martina.catalano@unifi.it (M.C.); valeriaemma.palmieri@unifi.it (V.E.P.); gianmarco.vannini@unifi.it (G.V.); eleonora.buttitta@unifi.it (E.B.); 2Histopathology and Molecular Diagnostics, Careggi Teaching Hospital, 50139 Florence, Italy; santir@aou-careggi.toscana.it (R.S.); galliic@aou-careggi.toscana.it (I.C.G.); 3Department of Experimental and Clinical Medicine, University of Florence, 50139 Florence, Italy; donata.villari@unifi.it; 4Clinical Oncology Unit, Careggi Teaching Hospital, 50139 Florence, Italy; rossiv@aou-careggi.toscana.it

**Keywords:** urothelial carcinoma, bladder cancer, PD-1, PD-L1, immune checkpoint blockade, biomarkers

## Abstract

**Simple Summary:**

Urothelial bladder cancer (BC) is one of the most fatal cancers, with a dismal five-year survival rate of 5% in patients with metastatic disease. Clinically relevant benefits of immunotherapy in advanced or metastatic bladder cancer have led to Food and Drug Administration (FDA) approval of immune checkpoint inhibitors (ICIs) as second- or first-line therapy in patients unresponsive to or ineligible for standard treatment. The advantage of ICIs is being investigated in various stages of BC, either as monotherapy or in combination with other drugs. In this review we discuss the role of ICIs in BC, highlighting their current clinical application and outlining future therapeutic perspectives.

**Abstract:**

Bladder cancer (BC) is the most common malignancy of the genitourinary tract, with high morbidity and mortality rates. Until recently, the treatment of locally advanced or metastatic urothelial BC was based on the use of chemotherapy alone. Since 2016, five immune checkpoint inhibitors (ICIs) have been approved by the Food and Drug Administration (FDA) in different settings, i.e., first-line, maintenance and second-line treatment, while several trials are still ongoing in the perioperative context. Lately, pembrolizumab, a programmed death-1 (PD-1) inhibitor, has been approved for Bacillus Calmette–Guérin (BCG)-unresponsive high-risk non-muscle invasive bladder cancer (NMIBC), using immunotherapy at an early stage of the disease. This review investigates the current state and future perspectives of immunotherapy in BC, focusing on the rationale and results of combining immunotherapy with other therapeutic strategies.

## 1. Introduction

Bladder cancer (BC) is the ninth-most common malignancy worldwide, with 83,730 estimated new cases in the USA in 2021 [1] and the seventh-most common cancer in men [2]. Tobacco smoke appears to be the most common risk factor for BC, accounting for approximately 50% of cases [3]. Compared with never smokers, BC risk is three-fold higher in former smokers and over six-fold higher in current smokers, steadily increasing with the number of cigarettes and years smoked [4]. Occupational exposure is responsible for 5–6% of urothelial carcinomas. Among dietary factors, alcohol appears to play a role in the pathogenesis of BC, while the intake of Vitamin D and daily consumption of fruit and vegetables could have a protective effect [5].

At the time of diagnosis, approximately 70% of urothelial carcinomas are superficial, while 30% present with muscle infiltration [2]. Treatment of non-muscle invasive bladder cancer (NMIBC) involves transurethral resection of the bladder tumor (TURBT) followed by intravesical chemotherapy or immunotherapy. Bacillus Calmette–Guérin (BCG) immunotherapy is the gold standard adjuvant treatment for NMIBC with a high risk of progression and is also recommended for intermediate-risk NMIBC [6].

The standard treatment for nonmetastatic muscle invasive bladder cancer (MIBC) (T2–T4, N0, M0) is neoadjuvant cisplatin-based therapy, succeeded by radical cystectomy (RC) and pelvic lymphadenectomy [7]. Patients undergoing RC for MIBC have a high risk of relapse, especially in cases of ≥pT2 disease and/or pathological lymph node involvement. Adjuvant cisplatin-based multi-chemotherapy may be considered for patients fulfilling platinum eligibility criteria that include at least one of the following: Eastern Cooperative Oncology Group (ECOG) performance status of 2, creatinine clearance less than 60 mL/min, grade ≥ 2 hearing loss, grade ≥ 2 neuropathy, and/or New York Heart Association Class III heart failure [8,9].

Cisplatin-containing chemotherapy is the preferred first-line treatment also in metastatic disease. The most commonly used regimens in this setting include a combination of gemcitabine and cisplatin (GC), methotrexate, vincristine, adriamycin and cisplatin (MVAC) every four weeks, or dose-dense (dd) MVAC every two weeks. The median overall survival (OS) rates are 13.8 months, 14.8 months, and 15.5 months for GC, MVAC, and ddMVAC regimens, respectively [10,11]. Outcome is poor for patients who are unfit for platinum chemotherapy or undergo progression after frontline platinum chemotherapy, however, a major milestone in the metastatic setting was the approval of immune checkpoint inhibitors (ICIs) (Table 1).

ICIs are monoclonal antibodies directed against cytotoxic T-lymphocyte–associated antigen 4 (CTLA-4), programmed death 1 (PD-1) receptor and programmed death ligand-1 (PD-L1). CTLA-4 is a membrane receptor acting as a major negative regulator of T cell responses through interaction with its ligands, CD80 (B7-1) and CD86 (B7-2), expressed on the surface of antigen-presenting cells. PD-1 is a membrane receptor expressed by T cells, particularly in conditions of chronic antigen exposure, and exerts an inhibitory action on lymphocytes by binding to its two ligands, PD-L1 and PD-L2. PD-L1 is expressed on immune cells, such as T cells, B cells, dendritic cells (DCs) and macrophages [20,21], while PD-L2 is expressed mainly on antigen-presenting cells (APCs), including macrophages and myeloid DCs [22,23]. PD-L1 and PD-L2 have differential functions in immune regulatory processes. Indeed, PD-L1 inhibits T cells in peripheral tissues, whereas PD-L2 suppresses immune T cell activation in lymphoid organs. PD-L2 also inhibits type 2 T-helper (T_H_2) lymphocytes, but its role is yet to be fully understood [24,25]. By interrupting the ligand/receptor interactions, the anti-CTLA-4 (ipilimumab, tremelimumab) and anti-PD-1 (nivolumab, pembrolizumab)/anti-PD-L1 (atezolizumab, durvalumab, avelumab) antibodies remove T cell inhibition, thus favoring antitumor cytotoxic activity [26] (Figure 1). Characterization of immune checkpoints has furthered development of novel immunotherapeutic agents with clinical activity against a variety of solid tumors, including BC.

This article reviews current evidence supporting the use of new checkpoint inhibitors in BC, along with information on biomarkers that may predict response to immunotherapy.

## 2. Non-Muscle Invasive Bladder Cancer (NMIBC)

In approximately 75% of BC patients, the disease is confined to the mucosa (stage Ta, carcinoma in situ) or submucosa (stage T1) [27]. Although TURB alone can eradicate TaT1 tumors completely, they commonly recur and can progress to MIBC, thus necessitating the use of adjuvant treatment. In patients with intermediate-risk tumors, one-year full-dose BCG treatment or chemotherapy instillations for a maximum of one year is recommended. Conversely, full-dose intravesical BCG for one to three years is indicated in patients with high-risk tumors [28].

Therapeutic options for patients with BCG-unresponsive disease include RC, further intravesical therapy, and systemic therapy. A relatively new addition to the landscape of treatment for BCG-unresponsive NMIBC is pembrolizumab. Initial results of the KEYNOTE-057 phase II trial were reported in February 2019 showing a 38.8% (40/102) complete response (CR) rate at 3 months. Following the presentation of these data, pembrolizumab received FDA approval in January 2020 for BCG-unresponsive high-risk NMIBC patients, ineligible for, or refusing RC. Key secondary endpoints were duration of response (DOR) and safety. At a median follow-up of 14 months, 72.5% of patients maintained CR, 25.0% experienced recurrent NMIBC after CR, but none progressed to MIBC. Treatment-related adverse events (AEs) occurred in 63.1% of patients, the most frequent being pruritus, fatigue, diarrhea, hypothyroidism, and maculopapular rash. Grade 3–4 AEs occurred in 12.6% of patients, and one death due to colitis was considered treatment-related [29]. Updated data over a 2-year follow-up were submitted at the 2020 American Society of Clinical Oncology (ASCO) Annual Meeting. The median DOR was 16.2 months, and CR rate was 40.6% with 46.2% of responses longer than 12 months. The median PFS and OS were not reached [30].

At the 2021 ASCO Genitourinary Cancers Symposium, Balar et al. reported additional results with an extended minimum follow-up of 26.3 months [31]. Among those patients achieving CR, 33.3% remained in CR for ≥18 months and 23.1% for ≥24 months as of the data cutoff date. Of the 41.7% patients undergoing cystectomy after discontinuation of pembrolizumab, 35 (88%) had no pathological upstaging to MIBC, three (8%) had evidence of MIBC, and two (5%) had no available pathology data. Safety profile remained consistent with what had been previously reported.

Another phase II trial, SWOG S1605, tested atezolizumab in the same setting. The primary outcome was the pathological complete response (pCR) rate at six months, accomplished through mandatory biopsy. A pCR was observed in 30 (41.1%) patients at 3 months and in 19 (26.0%) at 6 months. The most common AEs were fatigue, pruritus, hypothyroidism, and nausea. Grade 3–5 AEs occurred in 12.3% of patients, and there was one treatment-related death due to myasthenia gravis [32].

Several clinical trials with other ICI agents, both as monotherapy and as part of a combination therapy, are ongoing and in early-stage BC. Particularly relevant are the POTOMAC trial assessing durvalumab plus BCG in BCG-naïve patients, the KEYNOTE-676 study evaluating BCG-associated pembrolizumab in patients with recurrence after induction BCG therapy alone [33], and the NCT03317158 trial establishing the safety of durvalumab as monotherapy and in combination with BCG and external beam radiation therapy (EBRT) in BCG-unresponsive NMIBC patients (Table 2).

## 3. Muscle Invasive Bladder Cancer (MIBC)

RC is the treatment of choice for MIBC, nevertheless approximately half of the patients are susceptible to high rates of local and distant relapse, potentially due to undetected occult micrometastases. Neoadjuvant and adjuvant cisplatin-based chemotherapy can be administered, offering modest survival benefit [34,35], however, not all patients are eligible for cisplatin on account of their age or comorbidities [36]. In early phase clinical trials, perioperative ICI strategies have shown encouraging outcomes, although results of phase III randomized controlled trials are eagerly awaited, and sensitive biomarkers are needed to support treatment decision.

### 3.1. Neoadjuvant Setting

Several anti-PD1, anti-PD-L1 and anti-CTLA-4 agents, either individually or in combination, had been investigated as neoadjuvant treatments prior to RC.

In two single-arm phase II trials (PURE-01 and ABACUS), pembrolizumab and atezolizumab as single agents have been tested in the neoadjuvant setting. In the PURE-01 study, patients with MIBC, regardless of cisplatin eligibility, were selected to receive three cycles of pembrolizumab before surgery, switching to chemotherapy (ddMVAC) in case of disease progression. The pCR rate was 42% and downstaging to <pT2N0 was achieved in 27 (54%) patients [37]. The ABACUS trial assessed the administration of two cycles of atezolizumab in cisplatin-ineligible patients. At a median follow-up of 13.1 months, the pCR rate and 1-year relapse-free survival (RFS) were 31% and 79%, respectively [38].

Data from a single-arm trial, combining durvalumab with tremelimumab (CTLA-4 inhibitor) in high-risk cisplatin-ineligible patients, were reported by Gao et al. [39]. In this study, the pCR was 49% and grade 3 immune-related AEs were observed in 17% of patients.

In another randomized phase II trial, the same ICI combination was compared with neoadjuvant cisplatin-based chemotherapy (GC or ddMVAC) in patients with urothelial MIBC (cT2-T4a N ≤ 1 M0), classified as “hot” or “cold” according to tumor immune score (TIS) [40]. Patients with “hot” tumors were randomized to durvalumab plus tremelimumab or standard chemotherapy. The pCR was 36.4% in the immunotherapy arm vs. 34.8% in the standard chemotherapy arm (95% CI, 0.26–3.24). Grade 3–4 AEs were more frequent in the chemotherapy arm.

A recent single-arm feasibility trial enrolled patients, either cisplatin-ineligible or refusing cisplatin-based chemotherapy, to receive nivolumab and the CTLA-4 inhibitor, ipilimumab, with a pCR of 45% [41]. Grade 3–4 immune-related AEs occurred in 55% of patients. Contrary to studies with anti-PD-1/PD-L1 monotherapy, CR to ipilimumab plus nivolumab was independent from baseline CD8+ presence or T-effector signatures.

A combination strategy with ICIs and PARP inhibitors was investigated in a phase II clinical trial. This study considered durvalumab plus olaparib in patients with cT2-T4a N0 M0 urothelial carcinoma, demonstrating a pCR rate of 44.5%, and grade 3–4 AEs in 8.3% of patients [42]. Evidence suggests that chemotherapy and immunotherapy have synergistic effects, therefore this approach may be associated with improved clinical outcomes.

In a phase Ib/II trial, neoadjuvant pembrolizumab combined with GC or gemcitabine in cisplatin-eligible or ineligible patients with cT2-T4a N0 bladder UC has been evaluated. In the cisplatin-eligible cohort, pCR was 44.4%, regardless of baseline PD-L1 score, with an estimated 3-year RFS and OS of 63% and 82%, respectively. The safety profile documented grade 3–4 cytopenia in 57% of patients [43]. In the cisplatin-ineligible cohort, the pCR, 1-year RFS and OS were 45.2%, 67% and 88.4%, respectively [44]. Nivolumab in combination with GC prior to RC was tested in the BLASST-1 trial where pCR occurred in 49% (20/41) of cases. The overall rate of grade 3–4 AEs was 24%, the majority being from GC [45].

Perioperative chemoimmunotherapy has been assessed in a phase II trial using four cycles of durvalumab in combination with GC followed by RC and adjuvant durvalumab. No tumor progression was recorded at preoperative restaging with a pCR achieved in 30% of patients, and a pathological response rate of 50%. Postoperative complications arose in 27% of cases, and infections were the most common (17%) [46].

Finally, the RETAIN BLADDER trial has been designed to evaluate a risk-adapted approach to the treatment of MIBC following neoadjuvant accelerated methotrexate, vinblastine, doxorubicin, and cisplatin (AMVAC) chemotherapy. Based on the mutational profile and the post-AMVAC biopsy findings, patients are being treated with active surveillance (experimental arm), or standard of care (SoC) intravesicle therapy, chemoradiation or surgery. Achievement of the endpoint (metastasis-free survival at 2 years) would preserve the bladder and improve quality-of-life for a proportion of patients (NCT02710734).

Overall, a combination of durable clinical activity and tolerability has expanded the utility of this group of drugs to early-stage disease, either as monotherapy or in combination with other agents. Given their favorable toxicity profile, ICIs may also provide benefits to a larger patient population, including those with impaired renal function owing to disease status and comorbidities. However, optimal timing between immunotherapy and surgery, number of cycles of ICIs before surgery, and continuation of treatment after RC must be clarified. Validation of predictive biomarkers to identify which patients to treat is a compelling clinical need.

### 3.2. Adjuvant Setting

The addition of systemic therapy following surgery, or adjuvant therapy, is regularly used in many solid tumors, but is not standard management in BC, largely owing to lack of clinical data and cisplatin-based chemotherapy ineligibility [47]. The low rate of treatment-related toxicities associated with ICIs make these agents an attractive therapeutic option in the postoperative setting.

The role of adjuvant ICIs in urothelial cancer (UC) was evaluated in three phase III clinical trials. Patients at high risk of recurrence were defined as those who had received neoadjuvant cisplatin-based chemotherapy with at least pathological T2 or node-positive disease, or those who had not received neoadjuvant chemotherapy with at least pathological T3 disease and were ineligible for cisplatin-based chemotherapy. The experimental arms consisted of atezolizumab, nivolumab or pembrolizumab given as single agents for up to 12 months. The IMvigor010 trial failed to meet its primary endpoint of improved disease-free survival (DFS) in the atezolizumab group recording, at a median follow-up of 21.9 months, a median DFS of 19.4 months vs. 16.6 months in the observation arm (HR, 0.89; 95% CI, 0.74–1.08; *p* = 0.24). Although relatively immature, these data do not support the use of atezolizumab in the adjuvant setting [48]. Conversely, the interim analysis of CheckMate-274, presented at the 2021 ASCO Genitourinary Cancers Symposium, showed a median DFS in all randomized patients of 21.0 months in the nivolumab arm vs. 10.9 months in the placebo arm (HR, 0.70; *p* = 0.0006) [49]. Finally, pembrolizumab is still being tested in a phase III randomized clinical trial, enrolling and randomizing patients to receive immunotherapy or observation alone in the adjuvant setting (NCT03244384).

## 4. Advanced or Metastatic Bladder Cancer

### 4.1. First-Line Therapy

The first-line treatment of metastatic urothelial cancer (mUC) is usually cisplatin-based combination chemotherapy [50,51,52]. However, despite high initial response rates, nearly all patients progress and die from BC. In addition, a subgroup of patients who are candidates for combination chemotherapy are unable to receive cisplatin due to renal dysfunction, neuropathy, severe hearing loss or heart failure.

The efficacy of carboplatin-based therapy was evaluated in the EORTC trial 30986, where 238 chemotherapy-naïve patients with impaired renal function (glomerular filtration rate <60 but >30 mL/min), and/or an Eastern Cooperative Oncology Group (ECOG) PS ≥2, were randomly assigned to treatment with carboplatin and gemcitabine, or methotrexate, carboplatin and vinblastine (MCAVI) [53]. The combination of gemcitabine and carboplatin proved to be as effective as MCAVI, with a better toxicity profile, therefore supporting use in patients with impaired renal function or a poor ECOG -PS ≥2 who are otherwise candidates for combination chemotherapy.

In recent years, ICIs have become an important therapeutic strategy in many solid tumors [54,55,56,57,58,59]. High levels of PD-L1 expression have been found in mUC with poor outcome. ICIs can be used to treat patients who are ineligible for any platinum-based (cisplatin or carboplatin) chemotherapy, regardless of PD-L1 expression status.

The efficacy of pembrolizumab as first-line therapy in mUC was evaluated in the phase II KEYNOTE-052 study, where 370 patients with advanced UC ineligible for a cisplatin-based regimen were given pembrolizumab at 200 mg every 21 days for up to 24 months [60]. After a minimum follow-up of two years, the objective response rate (ORR) was 29% for the entire cohort, comprising 9% CR and 20% partial response (PR). The median DOR and OS was 30 and 11.3 months, respectively [18]. The ORR was higher in patients with PD-L1 expression >10%, but responses were also observed in those with PD-L1 expression <10%.

The phase III KEYNOTE-361 trial is a randomized study comparing (1:1:1) pembrolizumab +/− chemotherapy (cisplatin or carboplatin plus gemcitabine) in 1010 patients with treatment-naïve unresectable or metastatic BC [61]. The median PFS and median OS were 8.3 and 17 months, respectively, for the combination therapy vs. 7.1 and 14.3 months for chemotherapy alone. This study failed to meet the co-primary endpoints of PFS and OS.

The DANUBE trial, a phase III study, compared durvalumab +/− tremelimumab vs. gold standard chemotherapy in patients with untreated unresectable locally advanced or metastatic BC [62]. This study enrolled 1032 patients distributed randomly (1:1:1) to receive durvalumab alone, durvalumab plus tremelimumab or chemotherapy. The two primary endpoints were OS between the groups treated with durvalumab alone and standard chemotherapy in the population with high PD-L1 expression, and OS between the durvalumab plus tremelimumab and SoC in the intention to treat (ITT) population. This trial failed to achieve either primary endpoint. After a median follow-up of 41.2 months, median OS in the durvalumab group was 14.4 vs. 12.1 months in the chemotherapy arm, while in the ITT population, median OS was 15.1 months in the experimental arm and 12.1 months in the chemotherapy group. The AEs grade 3 or 4 were far more severe and frequent in patients receiving chemotherapy (60%) than in patients treated with durvalumab (14%) or durvalumab plus tremelimumab (27%) [62].

The efficacy of durvalumab is also being investigated in the NILE trial, a multicenter phase III study of 1215 patients with locally advanced or metastatic BC, who were randomized to receive durvalumab +/− tremelimumab with platinum-based chemotherapy (1:1:1). The original co-primary endpoints were PFS and OS for durvalumab plus chemotherapy vs. chemotherapy in the ITT population [63].

Atezolizumab was approved by the FDA based on the results of the IMvigor210 trial, testing atezolizumab as first-line therapy in patients with cisplatin-ineligible mUC [14]. PD-L1 expression on tumor-infiltrating immune cells (ICs) was assessed by immunohistochemistry, with categories defined by percentage of positive cells: IC0 (<1%), IC1 (≥1% but <5%), and IC2/3 (≥5%) [12]. After a median follow-up of 17 months, OR was observed in 23% (95% CI, 16–31) of patients: 28% in IC2/3, 24% in IC1/2/3, 21% in IC1, and 21% in IC0 patients. Median DOR was not reached, and 19 of 27 responses were ongoing at the time of analysis. Median PFS was 2.7 months (95% CI 2.1–4.2) in all patients, 4.1 months in IC2/3 patients, 2.1 months in IC1 patients, and 2.6 months in IC0 patients. Median OS was 15.9 months (95% CI, 10.4 to not estimable) for the whole cohort, 12.3 months for IC2/3 patients, and 19.1 months in IC0/1 patients.

The most important trial investigating atezolizumab as first-line therapy is the IMvigor130 study, a placebo-controlled phase III trial in patients with untreated locally advanced or metastatic BC. It consisted of 1213 patients randomized into three groups: atezolizumab plus platinum-based chemotherapy, atezolizumab as monotherapy, and platinum-based chemotherapy plus placebo. After a median follow-up of 11.8 months, the median PFS was 8.2 months in the atezolizumab plus chemotherapy group vs. 6.3 months in the placebo plus chemotherapy group (HR, 0.82; 95% CI, 0.70–0.96; *p* = 0.007) [64].

### 4.2. Maintenance Therapy

In mUC patients, maintenance therapy with ICIs can be administered after an objective response (OR), or disease stability, to platinum-based chemotherapy regardless of PD-L1 tumor status.

The JAVELIN Bladder 100 study is a randomized phase III trial enrolling 700 patients with locally advanced or metastatic BC who showed OR, CR or partial response (PR), or else stable disease (SD), after four to six cycles of platinum-containing chemotherapy [19]. The patients were randomly assigned to either maintenance avelumab plus best supportive care (BSC) or BSC alone (control). OS at 1 year reached 71.3% in the avelumab group and 58.4% in the control group (*p* = 0.001). Avelumab also significantly extended OS in the PD-L1-positive population, with a 1-year OS of 79.1% in the avelumab group and 60.4% in the control group (*p* < 0.001). AEs from any cause reached 98.0% in the avelumab group and 77.7% in the control group, with grade 3 or higher AEs occurring in 47.4% and 25.2% of patients, respectively. Based on these results, the US FDA approved avelumab for maintenance therapy in patients with locally advanced or metastatic BC, not progressing on initial platinum-based chemotherapy.

Hoosier Cancer Research Network’s GU14-182 is a phase II trial enrolling patients randomly assigned to receive maintenance placebo or pembrolizumab after platinum-based chemotherapy. Most patients had visceral metastatic disease. The primary endpoint was to define PFS in accordance with immune-related response evaluation criteria in solid tumors (RECIST). PFS was significantly longer with pembrolizumab than with the placebo (5.4 months and 3.0 months, respectively). Median OS was 22 months (95% CI, 12.9 months to not reached) with pembrolizumab, and 18.7 months (95% CI, 11.4 months to not reached) with the placebo. No significant interaction was found between PD-L1 combined positive score (CPS) ≥ 10 and treatment arm for PFS and OS [65].

The role of other agents has not yet been established in the maintenance setting. Indeed, vinflunine, a third-generation bifluorinated semi-synthetic vinca alkaloid, has shown progression-free, but not OS benefit in randomized phase II trials [66].

### 4.3. Second-Line Therapy and Beyond

Over the last few years, two PD-1 (nivolumab and pembrolizumab) and three PD-L1 (atezolizumab, durvalumab and avelumab) inhibitors have been approved in the metastatic setting [67].

Based on the KEYNOTE-045 study, pembrolizumab was approved by the FDA in May 2017 for patients with locally advanced or metastatic disease which recurred or progressed after platinum-containing chemotherapy [17]. Pembrolizumab was associated with significantly longer OS (10.3 months vs. 7.4 months in the chemotherapy group), with fewer treatment-related AEs.

Following the results of the phase II trial IMvigor210, atezolizumab was approved by the FDA in May 2016 for the treatment of locally advanced or metastatic BC after failure of platinum-based chemotherapy [12]. In this study, 310 patients with advanced disease were enrolled to receive atezolizumab. At a median follow-up of 11.7 months, the ORR was 15%, with a manageable safety profile of the drug. Increased levels of PD-L1 expression on immune cells (>5% of tumor-infiltrating lymphocytes expressing PD-L1 determined by immunohistochemistry) were associated with increased response. Grade 3–4 treatment-related AEs occurred in 50 (16%) patients while grade 3–4 immune-mediated AEs in 15 (5%), with prevalence of pneumonitis, increased aspartate aminotransferase (AST) and alanine aminotransferase (ALT), rash, and dyspnea. No treatment-related deaths occurred during the study.

Safety and efficacy of atezolizumab vs. chemotherapy (docetaxel, paclitaxel or vinflunine) were assessed in the phase III IMvigor211 trial, where 931 patients were assigned and received atezolizumab or chemotherapy. Atezolizumab was not associated with significantly longer OS than chemotherapy, but its safety profile was more favorable. Grade 3–4 treatment-related AEs were lower in the atezolizumab group (20%) than in the chemotherapy group (43%) [68].

In mUC, safety and antitumor activity of nivolumab were first recorded in the nonrandomized CheckMate-032 study that assessed the efficacy of nivolumab, alone or in combination with ipilimumab, in several advanced tumor settings. OR was achieved in 19 of 78 (24.4%) mUC patients of the nivolumab monotherapy group (95% CI, 15.3–35.4) with a manageable safety profile [69]. In this study, patient response did not appear to be influenced by PD-L1 expression on tumor cells.

The single-arm, phase II CheckMate-275 study evaluated nivolumab in 270 patients with locally advanced, surgically unresectable or metastatic disease, which had progressed despite previous platinum-containing therapy [13]. Median follow-up for OS was 7 months (interquartile range, 2.96–8.77) and confirmed OR was achieved in 52 of 265 patients, regardless of PD-L1 expression (95% CI, 15.0–24.9). Grade 3–4 treatment-related AEs occurred in 48 (18%) patients, usually grade 3 fatigue and diarrhea. Three deaths (pneumonitis, acute respiratory failure, and cardiovascular failure) were attributed to treatment. The favorable results of this study led to accelerated FDA approval of nivolumab for treatment of patients with locally advanced or mUC who have disease progression during or following platinum-based chemotherapy.

In a phase I/II multicenter, open-label study, durvalumab administered every 2 weeks showed favorable clinical activity in platinum-refractory UC patients [16]. The ORR was 17.8% (95% CI, 12.7–24.0), including 7 CR. Responses occurred early (median time to response, 1.41 months), were durable (median DOR not reached), and did not differ by PD-L1 expression status. Grade 3–4 treatment-related AEs were seen in 13 patients (6.8%) and grade 3–4 immune-mediated AEs in 4 patients (2.1%).

The JAVELIN Solid Tumor study, a phase Ib single-arm trial, assessed the safety and efficacy of avelumab in patients with refractory mUC [15]. Forty-four patients were treated with avelumab (10 mg/kg every 2 weeks) and followed for a median of 16.5 months. Median PFS was 11.6 weeks (95% CI, 6.1 to 17.4 weeks) and median OS 13.7 months (95% CI, 8.5 months to not estimable), with a 12-month OS rate of 54.3%. The most frequent treatment-related AEs of any grade were fatigue/asthenia (31.8%), infusion-related reaction (20.5%), and nausea (11.4%). Grade 3–4 asthenia and increase of AST and creatine phosphokinase (CPK) occurred in 3 (6.8%) patients.

## 5. Biochemical and Clinical Predictors of Response to Immune Checkpoint Inhibitors (ICIs)

Only a subset of mUC patients undergoing ICI therapy develop a concrete and lasting response. Indeed, some patients receive little or no benefit from immunotherapy. The mechanisms underlying the marked variability of response to ICIs are not yet fully known and it is crucial to identify predictive biomarkers as well as resistance mechanisms in ICI non-responders. Currently FDA-approved biomarkers are PD-L1 expression and microsatellite instability-high (MSI-H)/mismatch repair deficiency (dMMR) for tumor agnostic therapy. Other emerging predictive biomarkers include tumor mutational load (TMB), gene expression profiles (GEP), the cancer genome atlas (TCGA) profile, and tumor infiltrating lymphocytes (TIL) [70].

### 5.1. PD-L1

In mUC, a satisfactory response rate was observed in patients with high PD-L1 expression treated in second-line with pembrolizumab and durvalumab, or in first-line with avelumab. However, no correlation was found between PD-L1 expression and OS in patients receiving second-line nivolumab and pembrolizumab [13,15,16,17,66,68,69,70,71,72,73]. PD-L1 expression detected by immunohistochemistry does not necessarily correlate with response to treatment. Some studies have shown that the negative predictive value of the test is poor and does not allow clinical discrimination between responders and non-responders [16,60,73]. Indeed, there is presently no standardized method and definition of PD-L1 positivity. Although immunohistochemistry is used as a reference test, staining methods and scoring systems vary considerably, leading to conflicting results. Lastly, heterogeneity in PD-L1 expression poses a further problem.

### 5.2. Tumor Mutational Burden (TMB)

High TMB is a leading candidate biomarker of immunotherapy response in various malignancies. TMB is defined as the number of somatic mutations (e.g., single-nucleotide variant, nonsense, missense, and splice site mutations) per megabase of interrogated genomic sequence harbored by cancer cells. Metastatic BC is characterized by high rates of somatic mutations, particularly in the Lund genomically unstable (LGU) group and TCGA cluster II group.

In the IMvigor210 trial, response to atezolizumab was significantly higher in TCGA cluster II patients than in other subtypes [14,74,75,76]. Indeed, increasing the number of mutant proteins would generate antigenic peptides, thereby enhancing immunogenicity [77]. A significant association between TMB and response to immunotherapeutic drugs has been observed in several tumor types, including mUC [14,17]. Recently, the CheckMate-275 study showed improved ORR (31.9% vs. 17.4%, *p* = 0.002) and PFS (3.02 months vs. 1.87 months) in patients with high TMB [78]. However, the clinical application and predictive power of this biomarker need to be further elucidated, and a way to discriminate responsive from nonresponsive patients has yet to be codified [79]. Low TMB does not necessarily correlate with lack of immunotherapy response, nor does high TMB always identify those patients most likely to benefit from immune checkpoint blockade [80].

### 5.3. Microsatellite Instability (MSI)

Mutations in genes involved in the MMR pathway cause defective DNA damage repair, which can be a predictive biomarker for ICI response [81]. Tumors with dMMR will also have additional mutations in non-MSI regions throughout the genome, and consequently more neoantigens than those with intact MMR, making antitumor immune response more effective and response to ICIs more likely. MSI/dMMR is therefore a cogent predictive biomarker for immunotherapy in such tumors, regardless of histological origin. MMR mutations have been associated with higher response rates to anti-PD-1 and anti-PD-L1 agents, and longer survival than in patients with high TMB alone [70]. Contrariwise, the IMvigor210 study showed better response in patients with high TMB than with dMMR [82]. This apparent divergence in results could be explained by the lack of standardized data and the mutation variability of the genes involved in MMR.

### 5.4. Gene Expression Profiles (GEP)

Gene expression profiling has been proven useful as a predictive biomarker of response to ICI treatment [83,84,85,86]. Of the sequenced genes, interferon-gamma (IFN-γ) expression may lead to consistently better prediction of ICI therapy outcomes. IFN-γ is a key cytokine for adaptive and innate anticancer mechanisms of the immune system, although it is also involved in immune evasion strategies of tumor cells. In the KEYNOTE-052 trial, which evaluated the efficacy of first-line pembrolizumab in cisplatin-ineligible advanced UC, GEP score showed a significant correlation with ORR (*p* < 0.0001) and could offer improved predictive performance to ICI response [87].

Recently, Tang et al. identified four immunotypes of BC, referred to as C1–C4, based on gene expression profiles. C2 had the highest degree of immune cell infiltration, while C4 exhibited a “desert”-like phenotype deprived of CD8+ cells. They demonstrated that the C2 subtype showed better OS and was more sensitive to anti-PD-1 treatment than other subtypes [88].

### 5.5. Tumor Infiltrating Cytotoxic T Lymphocytes

High infiltration of T lymphocytes in the tumor microenvironment has been correlated to a higher response rate with increased DFS and OS. Although there is no conclusive evidence, a retrospective analysis of 31 patients with muscle-invasive UC found that patients with more than eight CD8+ TILs had a longer median survival than patients with less than eight CD8+ cells [89]. Some clinical trials, e.g., IMvigor210 and PCD4989g, have attempted to outline a preliminary prognostic profile of response to treatment in mUC [90]. It has been observed that the absence of visceral metastases and neutrophil-to-lymphocyte ratio (NLR) lower than 5 are associated with clinical benefits. Neutrophilia and a higher concentration of neutrophils within the tumor could impair T cell function through overexpression of PD-L1 [91]. The assessment of TILs in routine pathology reports for advanced UC patients may be helpful but warrants further research as a standardized and validated biomarker.

## 6. Combination Strategies with Immune Checkpoint Inhibitors (ICIs)

Chemotherapy remains a critical component of the treatment armamentarium of UC and is unlikely to be replaced by ICIs or new target therapies in the immediate future. However, chemotherapeutic agents may have a synergistic effect and amplify the activation of CD8+ T cells achieved with ICIs.

Immunosuppressive cells, such as myeloid-derived suppressor cells (MDSCs), can induce immune tolerance, impeding the recognition and destruction of BC cells [92]. MDSCs are recruited and activated by tumor-derived factors and directly promote tumor growth, neovascularization and metastasis. Migrating into the tumor microenvironment, MDSCs also inhibit the antitumor reactivity of T lymphocytes and natural killer (NK) cells. Several chemotherapeutic drugs have been reported to regulate the immune environment of cancers, among which cisplatin can selectively deplete granulocytic-MDSCs (G-MDSCs), thus maintaining the function of CD8+ T cells against cancer [93].

### 6.1. ICIs with Chemotherapy

Data from the IMvigor210 trial suggest that chemotherapy can modulate PD-L1 expression in the tumor microenvironment. High PD-L1 expression was associated with improved ORR in patients previously treated with chemotherapy (cohort 2), but not in patients who had not received chemotherapy (cohort 1) [12]. A retrospective study investigating the impact of neoadjuvant chemotherapy on PD-L1 expression in urothelial carcinoma showed that PD-L1 levels increased following chemotherapy [94].

With regard to PFS, the IMvigor130 trial showed the advantage of combined chemo-immunotherapy (atezolizumab) over chemotherapy alone (8.2 months vs. 6.3 months; HR, 0.82; 95% CI, 0.70–0.96; *p* = 0.007), while no statistically significant benefit on OS was seen at the interim analysis after a median follow-up of 11.8 months [64]. Similarly, the final analysis of the KEYNOTE-361 study suggests that the addition of pembrolizumab to first-line platinum-based chemotherapy does not yield survival benefits in patients with advanced UC [61].

### 6.2. ICI Combination

The combination of PD-1/PD-L1 inhibitors and anti-CTL-4 antibodies seems to increase antitumor activity [62]. In the CheckMate-032 phase I/II trial, patients who progressed after platinum-based chemotherapy received either nivolumab or one of two nivolumab plus ipilimumab combination regimens [95]. Patients were treated with nivolumab 3 mg/kg (NIVO3) every 2 weeks or nivolumab 3 mg/kg + ipilimumab 1 mg/kg (NIVO3 + IPI1) or nivolumab 1 mg/kg + ipilimumab 3 mg/kg (NIVO1 + IPI3) every 3 weeks for 4 cycles followed by nivolumab 3 mg/kg every 2 weeks. The ORR was 25.6%, 26.9%, and 38.0% in the NIVO3, NIVO3 + IPI1, and NIVO1 + IPI3 arms, respectively. Grade 3 or 4 treatment-related AEs occurred in 21 (26.9%), 32 (30.8%), and 36 (39.1%) patients receiving NIVO3, NIVO3 + IPI1, and NIVO1 + IPI3, respectively. These findings not only encourage further investigation into NIVO1 + IPI3 in mUC, they also demonstrate the added benefit of immunotherapy combinations in this disease [95].

### 6.3. ICIs with Target Therapies

Several studies on ICIs combined with target therapies are currently ongoing. The phase Ib/II clinical trial FORT-2 evaluates the use of rogaratinib, an oral pan-FGFR1-4 inhibitor, in combination with atezolizumab in patients with first-line cisplatin-ineligible, FGFR-positive, advanced/mUC. Preliminary results, presented at the 2020 ASCO meeting, showed an ORR of 39% and a disease control (DC) rate of 65% [96].

Clinical activity of vofatamab, an antibody against FGFR3 blocking activation of both the wild-type and genetically activated receptor, was assessed in combination with pembrolizumab in the phase Ib/II FIERCE-22 clinical trial [97]. ORR was 32%, however, further studies are mandatory to better elucidate the synergy between FGFR inhibition and checkpoint blockade immunotherapy.

### 6.4. ICI with Antiangiogenic Drugs

Angiogenesis is known to play a pivotal role in the natural history of various malignancies, contributing to the pathogenesis and progression [98]. ICIs plus antiangiogenic agents have been confirmed to improve clinical outcomes in metastatic kidney cancer and are also currently being evaluated in UC.

Final results from a phase I trial and expansion cohorts of cabozantinib and nivolumab alone or with ipilimumab for metastatic genitourinary tumors were presented at the 2021 ASCO Genitourinary Cancers Symposium. The ORR for UC was 42.4% with 21.2% of CR, and median OS was 24.9 months (95% CI, 11.8–41.6) [99]. These regimens also demonstrated manageable safety. Grade 3–4 treatment-related AEs, including fatigue, diarrhea, and hypertension, occurred in 75% and 87% of patients treated with cabozantinib and nivolumab (CaboNivo) or CaboNivo plus ipilimumab (CaboNivoIpi), respectively. Grade 3–4 immune-related AEs were only seen in the triplet, and encompassed hepatitis (13%) and colitis (7%) [100].

In the phase Ia/b JVDF trial enrolling patients with previously treated advanced UC, the combination of pembrolizumab with ramucirumab, an IgG1 monoclonal antibody that binds to the extracellular domain of vascular endothelial growth factor receptor-2 (VEGFR2), achieved confirmed OR in 3 (13%) of 24 patients [101]. Nevertheless, this approach needs to be investigated in future trials, either with or without chemotherapy.

### 6.5. ICIs with Antibody-Drug Conjugates

Antibody-drug conjugates (ADCs) comprise a tumor-specific monoclonal antibody conjugated to a potent cytotoxin via a chemical linker [102]. The chemotherapeutic agent is only released within those cells expressing the protein target after internalization of the ADC and lysosomal cleavage.

Enfortumab vedotin (ASG-22ME) is an ADC composed of an anti-nectin-4 monoclonal antibody attached to monomethyl auristatin E, a microtubule-disrupting agent. This agent targets nectin-4, a cell adhesion molecule highly expressed in numerous cancers, including UC [103]. EV-201 is a global, phase II, single-arm study of enfortumab vedotin for patients with locally advanced or metastatic BC, previously treated with platinum-containing chemotherapy and PD-1/L1 inhibitors. Enfortumab vedotin was administered to 125 mUC patients with an ORR of 44% (95% CI, 35.1–53.2), including 12% CR. The most common treatment-related AEs were fatigue (50%), peripheral neuropathy (50%), alopecia (49%), rash (48%), decreased appetite (44%), and dysgeusia (40%) [71]. Safety and anti-tumor activity of enfortumab vedotin, alone or in combination with pembrolizumab and/or chemotherapy, are being studied in the first-line setting (NCT03288545). Preliminary data from the 2020 ASCO Annual Meeting showed encouraging and durable activity of pembrolizumab plus enfortumab vedotin with an ORR of 73.3% and median PFS of 12.3 months [104]. Moreover, a tolerable and stable safety profile was reported in cisplatin-ineligible patients. Based on these results, in February 2020, the FDA granted breakthrough therapy designation for enfortumab vedotin combined with pembrolizumab as first-line treatment in cisplatin-ineligible patients with unresectable locally advanced or mUC. Updated data with 24.9-month median follow-up were presented at the 2021 ASCO Annual Meeting. Median DOR was 25.6 months and OS rate 56.3% (95% CI, 39.8–69.9), with a manageable safety profile [105].

Sacituzumab govitecan (SG) is an ADC directed against trophoblast cell surface antigen 2 (Trop-2), a transmembrane glycoprotein expressed on the surface of most epithelial cancer cells. SG activity is under investigation in mUC patients progressing after prior platinum and ICI therapies, and preliminary results are encouraging [106]. In 113 patients who received SG, the ORR was 27%, and median DOR was 7.2 months, with median PFS and OS of 5.4 and 10.9 months, respectively.

## 7. Conclusions

The introduction of ICIs has dramatically changed the treatment paradigm for locally advanced or metastatic BC. Since 2016, ICIs have shown clinical benefits with a significant impact on OS and a durable tumor control in first-line therapy or upon relapse after standard treatments. More recently, immune checkpoint blockade has also proven beneficial in early-stage disease. New combinatorial strategies are under investigation to improve UC management, while further biomarker development is required to guide treatment in individual patients.

## Figures and Tables

**Figure 1 cancers-13-04411-f001:**
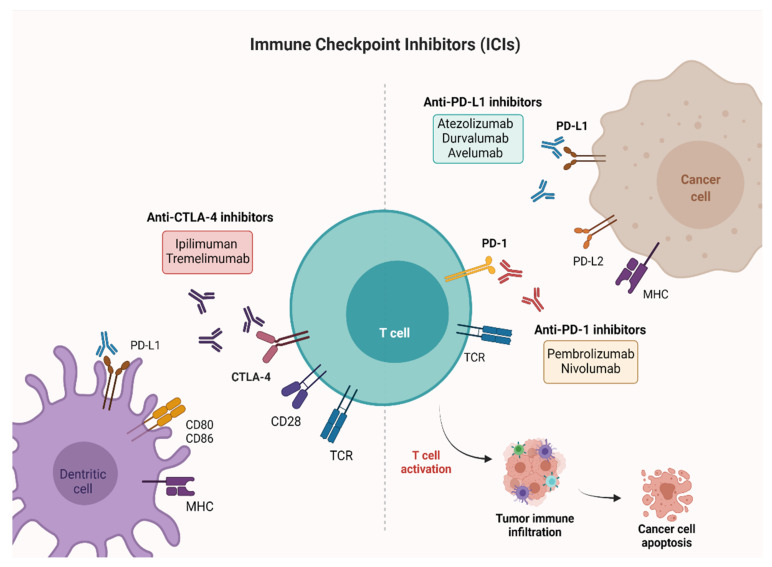
Mechanisms of action of ICIs targeting PD-1, PD-L1, and CTLA-4. PD-1 and CTLA-4 are proteins expressed on activated T cells. Their binding to the respective ligands presented on the surface of cancer cells leads to T cell inactivation and prevents tumor cell death. The immune checkpoint blockade ensures the activation of T cells and favors antitumor activity. Created with BioRender.com (accessed on 26 July 2021). PD-1: Programmed cell death-1; PD-L1: Programmed cell death-ligand 1; CTLA-4: Cytotoxic T-lymphocyte-associated antigen 4.

**Table 1 cancers-13-04411-t001:** Currently approved ICIs administered in urothelial bladder carcinoma.

Trial	Phase	FDA Approval	No. of Patients	ICI Therapy	Line of Treatment	Previous Platinum Therapy	Efficacy Outcomes
IMvigor210 [12]	II	May 2016	310	Atezolizumab	Second line	Yes	mPFS: 2.1 mo mOS: 7.9 mo ORR: 18%
CheckMate-275 [13]	II	February 2017	265	Nivolumab	Second line	Yes	mPFS: 2.0 mo mOS: 8.7 mo ORR: 20%
IMvigor210 [14]	II	April 2017	123	Atezolizumab	First line PD-L1+ platinum ineligible patients	No	mPFS: 2.7 mo mOS: 15.9 mo ORR: 23%
JAVELIN Solid Tumor [15]	I	May 2017	44	Avelumab	Second line	Yes	mPFS: 11.6 wk mOS: 13.7 mo ORR: 18.2%
Study 1108 [16]	I/II	May 2017	191	Durvalumab	Second line	Yes	mPFS: 1.5 mo mOS: 18.2 mo ORR: 18%
KEYNOTE-045 [17]	III	May 2017	542	Pembrolizumab	Second line	Yes	mPFS: 2.1 mo mOS: 10.3 mo ORR: 21%
KEYNOTE-052 [18]	II	May 2017	370	Pembrolizumab	First line PD-L1 + platinum ineligible patients	No	mPFS: 2.2 mo mOS: 11.3 mo ORR: 29%
JAVELIN Bladder 100 [19]	III	June 2020	700	Avelumab	Maintenance therapy	Yes	mPFS: 3.7 mo mOS: 21.4 mo

Immune checkpoint inhibitor (ICI); programmed death ligand 1 (PD-L1); median progression-free survival (mPFS); median overall survival (mOS); objective response rate (ORR); months (mo); weeks (wk).

**Table 2 cancers-13-04411-t002:** Ongoing phase II/III trials with active recruitment on ICIs alone or in combination with chemotherapy in different settings of BC treatment.

Trial	Phase	Allocation	No. of Patients	Study Populations	Line of Treatment	Experimental Arms	Primary Outcome
NCT02736266	II	N/A	90	MIBC	neoadjuvant prior to chemoradiation	Pembrolizumab	pCR
NCT02845323	II	randomized	44	MIBC	neoadjuvant	Nivolumab + Urelumab vs. Nivolumab	Immune response (tumor infiltrating CD8+ T cell density)
NCT03520491	II	not randomized	45	Cisplatin-ineligible patients with MIBC	neoadjuvant	Nivolumab and Nivolumab + Ipilimumab	No. of patients who proceed to RC-PLND
NCT03472274	II	randomized	99	BC patients	neoadjuvant	Durvalumab and Tremelimumab	Antitumor activity
NCT03732677	III	randomized	1050	MIBC	neoadjuvant/adjuvant	Durvalumab + Gemcitabine + Cisplatin neoadjuvant treatment followed by Durvalumab alone for adjuvant treatment	EFS
NCT04138628	II	randomized	282	Treatment of mBC at the time of biochemical relapse following RC	adjuvant	Atezolizumab	CR
NCT03244384	III	randomized	739	Locally advanced and mUC	adjuvant	Pembrolizumab vs. observation	OS, DFS
NCT04223856	III	randomized	760	Previously untreated locally advanced or mUC	1st	Enfortumab vedotin + Pembrolizumab vs. chemotherapy alone	PFS, OS
NCT03036098	III	randomized	1290	Unresectable or mUC	1st	Nivolumab + Ipilimumab, or SoC chemotherapy vs. SoC Chemotherapy	OS, PFS
NCT03682068	III	randomized	1434	Unresectable locally advanced or mUC	1st	Durvalumab + SoC chemotherapy and Durvalumab + Tremelimumab and SoC Chemotherapy vs. SoC chemotherapy alone	OS
NCT03898180	III	randomized	694	Locally advanced or mUC	1st	Pembrolizumab + Lenvatinib vs. Pembrolizumab +placebo	PFS, OS
NCT03697850	II	randomized	77	MIBC patients ineligible for RC	maintenance therapy	Atezolizumab	DFS

Not applicable (N/A); number (No.); metastatic urothelial cancer (mUC); metastatic bladder cancer (mBC); standard of care (SoC); best supportive care (BSC); overall survival (OS); progression free survival (PFS); disease free survival (DFS); events free survival (EFS); pathological complete response (pCR); radical cystectomy and pelvic lymph node dissection (RC-PLND); muscle-invasive bladder cancer (MIBC).

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
