# Peer review of "Immune Checkpoint Inhibitors in Urothelial Bladder Cancer: State of the Art and Future Perspectives"

_cancers, 2021, doi:10.3390/cancers13174411_

Round 1

Reviewer 1 Report

I commend the authors on a comprehensive overview of immune checkpoint inhibitors in urothelial cancers. The presented manuscript is clear and concise with its language, follows a logical progression and is well organized. Figures and tables are also well presented. The manuscript is highly relevant, and even includes recently presented studies at this year’s ASCO conference. This manuscript is of value, and will be an excellent resource on the up to date literature on IO in UC. With very few revisions, this manuscript is acceptable for publication in my opinion.

  • Introduction – “smoking alone causes 50% of bladder carcinoma’s”. Would re-phrase this to express the idea that smoking is the predominant or driving risk factor—not cause.
  • Introduction – may want to clearly define ‘cisplatin ineligibility’. It is surprising how often this criterion is not recognized in practice.
  • Section 2: NMIBC: ADAPT bladder is looking at IO in the BCG unresponsive setting. Should be mentioned.
  • Section 3: MIBC: onging trials are investingting a risk adapted approach, with potential for bladder sparing therapy. RETAIN 2 is ongoing, and employs IO. Risk adapted therapy should be mentioned.
  • Section 3.1, you have repeated three paragraphs inappropriately. Paragraphs starting with – 1) ‘A recent’, 2) ‘A combination’ 3)’ In a phase’. These need to be edited out of course.
  • Section 3.1, proof reading: com-pared. Please remove hyphen.
  • Section 4.1, IMVigor210 : please expand on data and responses with respect to TIL cut off.
  • Section 5.4: GEP paragraph is well written. Please expland on recent data from Tang et al (Front Oncology) on four immune subtypes in UC.
  • The manuscript encompasses only immune checkpoint inhibitors and not alternative immunotherapeutic’s (ie: immune modulators, vaccines, ect).Title may be misleading, please adjust to ICI instead of immunotherapy.
  • Section 6.5: ICI with ADC’s: Please make mention of the phase Ib/II EV-103 trial and the exciting topline results presented at last years ASCO. This may be a practice changing advance.

Author Response

Reviewer #1

I commend the authors on a comprehensive overview of immune checkpoint inhibitors in urothelial cancers. The presented manuscript is clear and concise with its language, follows a logical progression and is well organized. Figures and tables are also well presented. The manuscript is highly relevant, and even includes recently presented studies at this year’s ASCO conference. This manuscript is of value, and will be an excellent resource on the up to date literature on IO in UC. With very few revisions, this manuscript is acceptable for publication in my opinion.

Thank you for your positive comments.

Introduction – “smoking alone causes 50% of bladder carcinoma’s”. Would re-phrase this to express the idea that smoking is the predominant or driving risk factor—not cause.

The sentence has been revised based on your advice (Lines 39-40).

Introduction – may want to clearly define ‘cisplatin ineligibility’. It is surprising how often this criterion is not recognized in practice.

As suggested, cisplatin eligibility criteria have been added in the Introduction section (Lines 59-63).

Section 2: NMIBC: ADAPT bladder is looking at IO in the BCG unresponsive setting. Should be mentioned.

We agree with this and have mentioned the ADAPT-BLADDER trial in the manuscript (Lines 163-165).

Section 3: MIBC: ongoing trials are investigating a risk adapted approach, with potential for bladder sparing therapy. RETAIN 2 is ongoing, and employs IO. Risk adapted therapy should be mentioned.

The relevant trial has been included (Lines 340-347).

Section 3.1, you have repeated three paragraphs inappropriately. Paragraphs starting with – 1) ‘A recent’, 2) ‘A combination’ 3)’ In a phase’. These need to be edited out of course.

Thank you for pointing this out. The repeated paragraphs have been erased.

Section 3.1, proof reading: com-pared. Please remove hyphen.

The hyphen has been removed.

Section 4.1, IMVigor210: please expand on data and responses with respect to TIL cut off.

Thank you for your comment. Data and responses with respect to TIL cut off have been expanded according to the subgroups (Lines 448-456).

Section 5.4: GEP paragraph is well written. Please expand on recent data from Tang et al (Front Oncology) on four immune subtypes in UC.

As suggested, we have cited the article by Tan et al. Please see lines 639-643.

The manuscript encompasses only immune checkpoint inhibitors and not alternative immunotherapeutic’s (ie: immune modulators, vaccines, etc). Title may be misleading, please adjust to ICI instead of immunotherapy.

We have changed the title to “Immune Checkpoint Inhibitors in Urothelial Bladder Cancer: State of the Art and Future Perspectives” to avoid any misunderstanding.

Section 6.5: ICI with ADC’s: Please make mention of the phase Ib/II EV-103 trial and the exciting topline results presented at last years ASCO. This may be a practice changing advance. 

Thank you for your advice. The trial and data presented at ASCO 2020 and 2021 have been included (Lines 765-776).

Reviewer 2 Report

This is a timely overview of immunotherapy approaches in urothelial cancer. Different schedules and indications are being mentioned. A lot of data are being presented and I would suggest to add an overview table where the indications for immunotherapy could be indicated nowadays. For the methodology it could have been interesting to have a proper literature search in order not to miss any publications.

Author Response

Reviewer #2

This is a timely overview of immunotherapy approaches in urothelial cancer. Different schedules and indications are being mentioned. A lot of data are being presented and I would suggest to add an overview table where the indications for immunotherapy could be indicated nowadays. For the methodology it could have been interesting to have a proper literature search in order not to miss any publications.

Thank you for your comments. We agree with your suggestion and have included Table 1 which summarizes all currently approved ICIs and their clinical indications in bladder cancer. Although a proper literature search could have benefitted the manuscript, we performed a narrative review and not a systemic review with meta-analysis, and therefore cannot supply this information.

Reviewer 3 Report

Immunotherapy in Urothelial Bladder Cancer: State of the Art and Future Perspectives

In this manuscript, Roviello and colleagues proposes a review of the current literature on the therapeutic efficiency of immune checkpoint inhibitors (ICI) in urothelial bladder cancer (BC).

The three main sections (i.e. 3, 4 and 5) of this work give insights into the therapeutic options benefits/limitations of ICI alone or in combination with conventional therapies for non-muscle invasive bladder cancer (NMIBC), muscle invasive bladder cancer (MIBC) as well as advanced metastatic bladder cancer. The last part of the manuscript (i.e. section 6) is dedicated to the description of biomarkers potentially predicting ICI responses.

In view of the constant advances in the treatment of BC, any regular update on the latest clinical research in this field should be welcome. To my opinion, this is a well-structured manuscript which only requires minor modifications before publication, as discussed below.

Minor points

Lines 3-4: the word “Immunotherapy” in the title is misleading, as the reader expects to gain some insights into all possible immunotherapy strategies (i.e, active immunotherapy, adoptive T cell transfer, etc.) and not only ICI. Why are these other approaches not presented in this work? Have they proven to be inefficient in BC? If so, this point should be clarified and mentioned.

Line 64: the authors should better describe the expression and distribution pattern of PD-1 and its ligands. PD-L1 and PD-L2 are not only expressed by the tumour microenvironment, as suggested by the authors. This would also give them the opportunity to explain the physiological role of ICI, which is missing in the present manuscript.  

Table 2: Erdafitinib is not mentioned/explained in the main text. The authors should explain the rationale behind targeting the FGFR signalling pathway by this tyrosine kinase inhibitor in BC treatment.

Author Response

Reviewer #3

In this manuscript, Roviello and colleagues proposes a review of the current literature on the therapeutic efficiency of immune checkpoint inhibitors (ICI) in urothelial bladder cancer (BC).

The three main sections (i.e. 3, 4 and 5) of this work give insights into the therapeutic options benefits/limitations of ICI alone or in combination with conventional therapies for non-muscle invasive bladder cancer (NMIBC), muscle invasive bladder cancer (MIBC) as well as advanced metastatic bladder cancer. The last part of the manuscript (i.e. section 6) is dedicated to the description of biomarkers potentially predicting ICI responses.

In view of the constant advances in the treatment of BC, any regular update on the latest clinical research in this field should be welcome. To my opinion, this is a well-structured manuscript which only requires minor modifications before publication, as discussed below.

Thank you for your positive comments.

Minor points

Lines 3-4: the word “Immunotherapy” in the title is misleading, as the reader expects to gain some insights into all possible immunotherapy strategies (i.e. active immunotherapy, adoptive T cell transfer, etc.) and not only ICI. Why are these other approaches not presented in this work? Have they proven to be inefficient in BC? If so, this point should be clarified and mentioned.

While it would have been interesting to explore other aspects of immunotherapy, our work focused on ICIs only. We have changed the title to “Immune Checkpoint Inhibitors in Urothelial Bladder Cancer: State of the Art and Future Perspectives” to avoid any misunderstanding.

Line 64: the authors should better describe the expression and distribution pattern of PD-1 and its ligands. PD-L1 and PD-L2 are not only expressed by the tumour microenvironment, as suggested by the authors. This would also give them the opportunity to explain the physiological role of ICI, which is missing in the present manuscript.  

As suggested, we have detailed the expression and distribution pattern of PD-1 and its ligands. Please see lines 79-85.

Table 2: Erdafitinib is not mentioned/explained in the main text. The authors should explain the rationale behind targeting the FGFR signalling pathway by this tyrosine kinase inhibitor in BC treatment.

Thank you for pointing this out. In the NCT03390504 trial, erdafitinib is not combined but compared with ICIs and therefore references have been deleted from Table 2.